# Assessing the potential role of copper and cobalt in stimulating angiogenesis for tissue regeneration

**Elia Bosch-Rué, Leire Díez-Tercero, Raquel Rodríguez-González, Begoña María Bosch-Canals, Roman A. Perez** *

Bioengineering Institute of Technology (BIT), Universitat Internacional de Catalunya (UIC), Sant Cugat del Vallès, Barcelona, Spain

* rperezan@uic.es

**Data Availability Statement:** All relevant data are within the manuscript and its Supporting information files.

## Abstract

The use of copper ($Cu^{2+}$) and cobalt ($Co^{2+}$) has been described to stimulate blood vessel formation, a key process for the success of tissue regeneration. However, understanding how different concentrations of these ions affect cellular response is important to design scaffolds for their delivery to better fine tune the angiogenic response. On the one hand, gene expression analysis and the assessment of tubular formation structures with human umbilical vein endothelial cells (HUVEC) revealed that high concentrations (10μM) of $Cu^{2+}$ in early times and lower concentrations (0.1 and 1μM) at later times (day 7) enhanced angiogenic response. On the other hand, higher concentrations (25μM) of $Co^{2+}$ during all time course increased the angiogenic gene expression and 0.5, 5 and 25μM enhanced the ability to form tubular structures. To further explore synergistic effects combining both ions, the non-toxic concentrations were used simultaneously, although results showed an increased cell toxicity and no improvement of angiogenic response. These results provide useful information for the design of $Cu^{2+}$ or $Co^{2+}$ delivery scaffolds in order to release the appropriate concentration during time course for blood vessel stimulation.

## 1 Introduction

Tissue regeneration involves different sequential phases in which blood vessel formation is one of the essential phases to ensure a successful tissue formation. For instance, bone regeneration is a multi-step process involving different events such as inflammation of damaged zone, blood vessel formation (angiogenesis), recruitment of osteoprogenitor cells to the zone of healing and their subsequent osteogenic differentiation for bone formation [1]. Bone has a native capacity to regenerate itself in situations in which the size of the defect is small, such as cracks and small fractures. However, when the size of the defect exceeds the critical size, bone has a limited capacity to regenerate itself [2]. Among the different events that take place in bone regeneration, blood vessel formation has been recognized as one of the key processes for the success of bone fracture healing, showing limited regeneration when the vascular network was poorly developed [3]. For this reason, bone defects exceeding $\geq 2.5$ cm are unable to

**Funding:** RAP was funded by the Government of Catalonia (2017 SGR 708), the Spanish Ministry (Ramón y Cajal fellowship (RYC2018-025977-I) and project RTI2018-096088-J-100 (MINECO/FEDER), and EBR, LDT, RRG predoctoral fellowship from Universitat Internacional de Catalunya (UIC). The funders had no role in study design, data collection and analysis, decision to publish, or preparation of the manuscript.

**Competing interests:** The authors have declared that no competing interests exist.

regenerate without aid [2], mainly due to lack of vascularization, which leads to necrosis and cell death. Vascularization is generally triggered by the appearance of growth factors that are naturally found in the tissues during regenerative processes and have been proved to be successful in angiogenesis stimulation, as long as a graft is present in the site of defect. In order to mimic the natural angiogenesis process that will eventually promote bone tissue regeneration, the use of synthetic grafts combined with growth factors, such as vascular endothelial growth factor (VEGF), have been used to stimulate blood vessel formation [4]. However, they have delicate handling properties and short half-life. An alternative approach to mimic native cellular functions is the use of ions, which have shown similar functionality as growth factors and that are as well intrinsically found in the human body at low concentrations or as trace elements [5, 6]. An advantage of ions compared to growth factors is their higher stability, allowing longer periods of storage and an easy manipulation [7]. Therefore, in the last years, ions have been considered as a promising alternative.

Among the biological active metal ions, copper ($Cu^{2+}$) and cobalt ($Co^{2+}$) have been proposed as good candidates for angiogenesis stimulation, as both ions are described to mimic hypoxic conditions by up-regulating hypoxia inducible factor-1α (HIF-1α). This, in turn, up-regulates angiogenic-related genes, such as vascular endothelial growth factor (VEGF), and therefore, triggers blood vessel formation [8, 9]. Despite their great potential, the doses released from synthetic grafts and hence administered into the site of defect should be in the therapeutic window while avoiding the induction of cytotoxicity. There are several strategies to incorporate ions within these synthetic grafts, allowing a fine tuning of their release upon implantation [10]. Despite the significant number of scaffolds that have incorporated these ions into their structure, the optimum concentrations required at each time point of the angiogenic response are still unknown. Generally, ion release from scaffolds is measured as a cumulative release after several days, obtaining the angiogenic response at the end point of the experiment [11, 12] or in a single time point during the course of the experiment [13–16], which is generally never in the initial hours [17–19]. Hence, it is difficult to understand and design novel scaffolds with optimum concentrations as well as with the adequate release patterns that will match the exact time points at which the different ions need to be present. For this purpose, tissue engineering and scaffold design have the need of systematic experimental plan to unravel these concentrations.

While the use of individual metal ions has proved to be beneficial for the stimulation of different cell functions, combinatorial approaches have demonstrated to induce synergistic effects. For instance, the combination of silicon ($Si^{4+}$) and strontium ($Sr^{2+}$) increased the expression of angiogenic genes, such as VEGF and KDR, in HUVEC cells *in vitro*, and significantly increased the percentage of blood vessel area formation *in vivo* [20]. In another study, authors demonstrated a significantly stronger stimulatory effect up-regulating angiogenic growth factors when $Cu^{2+}$ and $Si^{2+}$ were combined together, requiring lower doses of each one compared with their individual effect [21]. Therefore, the combination of ions might serve as another strategy to stimulate blood vessel formation.

In this study, we aimed to screen a range of concentrations of two hypoxia mimicking ions ($Cu^{2+}$ and $Co^{2+}$) in order to evaluate their non-toxic concentrations and, at the same time, to evaluate how different concentrations affects the angiogenic response during early (hours) and later (days) time points. Furthermore, the possible synergistic combinatorial effect of these non-toxic ion concentrations was assessed, in order to open the possibility of combining ions to stimulate angiogenesis as this has not been studied before. Overall, the results may provide insights of how to design tissue engineering scaffolds that release ions for the early control of angiogenesis.

## 2 Materials & methods

### 2.1. Cell culture

Human Umbilical Vein Endothelial Cells (HUVEC; Lonza) were used to assess the angiogenic response. For cell expansion, HUVEC were seeded at a density of 2500 cells/cm$^2$ and maintained with endothelial growth medium-2 bulletkit (EGM-2) (Lonza; Ref. H3CC-3162), containing VEGF, rhFGF-B, rhEGF, r-IGF-I, hydrocortisone, ascorbic acid, gentamicin sulfate, amphotericin-B and 2% FBS. When cells reached 70–85% confluence, they were passaged using 0.25% Trypsin-EDTA (Gibco). Cells were maintained in standard culture conditions (37˚C and 5% $CO_2$). Passages equal or below P5 were used for subsequent assays. Preliminary experiments tested two types of cell culture media supplemented with ions: i) basal medium (Lonza; Ref. CC-3156), supplemented with 2% FBS and amphotericin-B; ii) and EGM-2 medium, previously described.

### 2.2. Cell cytotoxicity assay

$CuCl_2$ and $CoCl_2$ (both from Sigma-Aldrich) were used as a source for cell culture media supplementation for $Cu^{2+}$ and $Co^{2+}$, respectively. For viability assay, concentrations of $Cu^{2+}$ tested were 0, 0.1, 1, 10, 100 and 200 μM and for $Co^{2+}$ concentrations assessed were 0, 0.5, 5, 25, 50 and 100 μM. HUVEC were seeded at a density of 5000 cells/cm$^2$. After 24h of seeding, the culture with ion supplemented medium started for a total of 7 days, with change of media every two days. Viability of cells was assessed using a commercially colorimetric available kit, specifically the Cell Counting Kit-8 (CCK-8) (Sigma-Aldrich). This kit contains a water-soluble tetrazolium salt WST-8 which is reduced by dehydrogenases of cells, resulting in a yellow-colored product (formazan). The amount of formazan generated is considered directly proportional to the number of viable cells, and has an absorbance spectrum peak at 450–460 nm. CCK-8 was performed according to manufacturer's instructions. Briefly, at time points of 2 and 7 days after medium supplementation with ions, CCK-8 solution was added in each well and incubated with cells for 3 hours. After this period, supernatant was placed in a 96 well plate (Greiner bio-one) and the absorbance was read at 450 nm using a multi-detection microplate reader (Synergy HT, BioTek). Results were normalized with the control medium, which did not contain supplemented ion. Percentages over 80% were considered non-toxic. The viability of HUVEC cells was also assessed with the combination of $Cu^{2+}$ and $Co^{2+}$ with the highest and half of the highest non-toxic concentrations that induced an angiogenic response.

### 2.3. Cell morphology with phalloidin

Cell morphology and organization was performed by immunofluorescence of actin filament staining with phalloidin (Acti-stain 488 fluorescent phalloidin, Cytoskeleton, Inc). For this purpose, 24 well plate coverslips were autoclaved and placed in each well. Then, 20 μL of 1% sterilized gelatin (Sigma-Aldrich) was added on the surface of each coverslip and incubated at room temperature for 30 min. Afterwards, the excess of gelatin was aspirated and 24 well plates were placed in the incubator at 37˚C for 30 minutes. After this incubation period, cell culture media was added and HUVEC were seeded in each well with a density of 5000 cells/cm$^2$. After 24h of seeding, the cell culture media was replaced with the one containing ion supplementation, with the concentrations that proved to be non-toxic. Control samples were cultured without ions supplementation. At the time points of 2 and 7 days after medium supplementation, cells were rinsed with PBS 1X and subsequently fixed with 4% PFA during 20 min at room temperature. Next, cells were rinsed three times with PBS 1X and permeabilized with 0.5% Triton-100x/PBS for 10 min. After three washes with PBS 1X, samples were incubated with 100

nM Acti-stain phalloidin (Cytoskeleton, Inc) for 30 min protected from light. Then, after PBS 1X rinsing, cells were incubated with DAPI (NucBlue Fixed Cell stain DAPI, LifeTechnologies) for 5 min. Finally, cells were rinsed one last time with PBS 1X and then, cover slips were detached from 24 well plate and mounted on a microscope slide with Fluoromount-G (Bio-Nova). Samples were kept in the dark at 4˚C and they were visualized by confocal laser microscopy (Leica SP8, LAS X software version 3.5.5.19976). Excitation/emission wavelengths were 480/520 nm for phalloidin and 405/460 for DAPI. Images were processed with the same confocal software (LAS X Life Science software, Leica).

## 2.4. Gene expression RT-qPCR

Gene expression was analyzed by quantitative real time polymerase chain reaction (qPCR) at early time points (1, 2, 4, 10 and 24h) and later time points (day 7 and 7) to assess the early and later angiogenic response, respectively. With the combination of ions, the gene expression was analyzed at day 2 and 7. Cell pellets were collected at time points above mentioned using 0.25% Trypsin-EDTA (Gibco) and spin centrifugation of 1500 rpm for 5 min. Then, total RNA was isolated from cells using NucleoSpin RNA Kit (Macherey-Nagel) including DNAse treatment step following manufacturer's instructions. Quantification of isolated RNA was performed using a microvolume plate (Take 3) to measure absorbance ratio of wavelengths 260/280 nm in a microplate reader (Synergy HT Multi-detection Microplate Reader, BioTek). A ratio of ~2 was considered pure RNA. Reverse-transcription (RT) of RNA to cDNA was performed using Transcriptor First Strand cDNA Synthesis Kit (Roche) following manufacturer's instructions using T100 Thermal Cycler (Bio-Rad). For the amplification and quantification of cDNA targets, QuantiNova SYBR Green PCR kit (Qiagen) was used following manufacturer's instructions. Briefly, 20 ng of cDNA per reaction were amplified under the following conditions: an initial heat activation step of 2 min at 95˚C followed by 40 cycles of denaturation for 5s at 95˚C and annealing/extension for 10s at 60˚C, using a CFX96 Real-Time PCR Detection System (Bio-Rad). The primers sequences used for vascular endothelial growth factor (VEGF), hypoxia-inducible factor 1-alpha (HIF-1$\alpha$), platelet endothelial cell adhesion molecule-1 (PECAM-1) and the endogen gen beta-actin (β-Actin) are listed in Table 1. To normalize data, β-Actin was used as internal reference for each reaction. Relative expression was calculated using the $2^{-\Delta\Delta Ct}$ method and all data of conditions tested were expressed as fold-changes compared to each time point control to better elucidate the angiogenic effects of the ions through time course.

## 2.5. Matrigel assay for tubular formation

*In vitro* tubular formation analysis was performed using Matrigel® Growth Factor Reduced (GFR) (Corning® Matrigel®, Ref. 734–0268) with HUVEC cells in 96 well plates. First, 75 μL

**Table 1. List of primers sequences used for cDNA amplification in RT-qPCR.**

|  | Gene | Primer | Sequence (5'-3') |
|---|---|---|---|
| **Endogen** | β-Actin | Forward | AGAGCTACGAGCTGCCTGAC |
|  |  | Reverse | AGCACTGTGTTGGCGTACAG |
| **Angiogenic** | VEGF | Forward | CCCACTGAGGAGTCCAACAT |
|  |  | Reverse | AAATGCTTTCTCCGCTCTGA |
|  | HIF-1$\alpha$ | Forward | TGCTCATCAGTTGCCACTTC |
|  |  | Reverse | TCCTCACACGCAAATAGCTG |
|  | PECAM-1 | Forward | TCAAATGATCCTGCGGTATTC |
|  |  | Reverse | CCACCACCTTACTTGACAGGA |

of Matrigel at 4˚C were transferred to each well and incubated for 10 min at room temperature. After this period, plates were incubated in a humidified incubator at 37˚C for 30 min. Then, 75 μL of cell culture media was added to each well. For the conditions of ion supplementation, the supplemented media contained doubled the final desired ion concentration (of non-toxic concentrations). Plates were incubated 30 min at 37˚C. Then, 70 000 cells/cm$^2$ were added in each well with a volume of 75 μL. At the time points of 4 and 8h, cells were rinsed once with PBS 1X and fixed with 4% PFA for 30 min. Afterwards, cells were rinsed three times and stored at 4˚C with PBS 1X. There were three triplicates per condition and two fields per well were photographed using an inverted microscope coupled with a camera (Nikon Eclipse TS2-S-SM) and its software (IC Measure software). The phase contrast images were analyzed using the freely Angiogenesis Analyzer plugin of Image J. Number of nodes, number of meshes and total tubule length were analyzed.

### 2.6. Statistical analysis

Statistical analysis was performed using SPSS software (SPSS v21, IBM). Kruskal-Wallis and Mann Whitney U non-parametric tests were used to compare viability, gene expression and tubule formation (number of nodes, meshes and total tubule length) between experimental groups and control of the same time point (data represented as mean ± standard deviation; n = 4 for viability and n = 3 for gene expression and Matrigel assay for each condition). Spearman non-parametric test was used to study the correlation between HIF-1α and VEGF gene expression. Control samples were cells without any supplementation ion in cell culture media.

## 3 Results

### 3.1. Viability, proliferation and morphology

**3.1.1. Copper (Cu$^{2+}$).**   In order to elucidate the ion concentrations which are not cytotoxic, HUVEC cells were cultivated under different copper (Cu$^{2+}$) and cobalt (Co$^{2+}$) concentrations. Additionally, their viability effect was tested with both basal and growth media, to find out which one was more appropriate to use for further assays. Generally, we could observe an increase of detached cells and cell debris (when HUVEC were cultivated with basal medium (BM) (S1 Fig) compared to growth medium (GM) (S2 Fig). Moreover, a reduction of cells was observed in a dose-dependent manner during culture time when Cu$^{2+}$ was added in both cell culture mediums, which was more evident with BM.

Results of viability assays were in accordance with the observations of cell culture. With BM, all the concentrations tested resulted to be cytotoxic at the end of culture (S3 Fig). Alternatively, when HUVEC were cultured with GM supplemented with Cu$^{2+}$, they presented a proper viability with Cu$^{2+}$ concentrations up to 10 μM (Fig 1A). Due to these viability results, we considered to use GM for the subsequent assays.

The influence of Cu$^{2+}$ on cell morphology and arrangement was evaluated with actin filament staining. At day 2, no apparent differences between control and experimental groups were observed (Fig 1B). From 2 to 7 days, cells reached confluence and started to grow and proliferate over the monolayer, and some cells acquired an elongated phenotype.

**3.1.2. Cobalt (Co$^{2+}$).**   Regarding HUVEC culture with BM supplemented with Co$^{2+}$, there were high amounts of cell debris and detached cells compared with GM (S4 **and** S5 **Figs**). These observations were more evident when the concentration of Co$^{2+}$ was increased, which was in accordance with viability results. With BM, all the concentrations resulted to be cytotoxic at the end of the experiment, (S6 Fig). Alternatively, when HUVEC cells were cultured with GM supplemented with Co$^{2+}$, concentrations up to 50 μM demonstrated to be not toxic

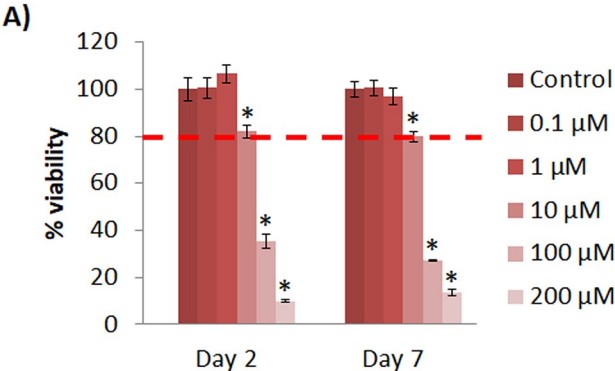

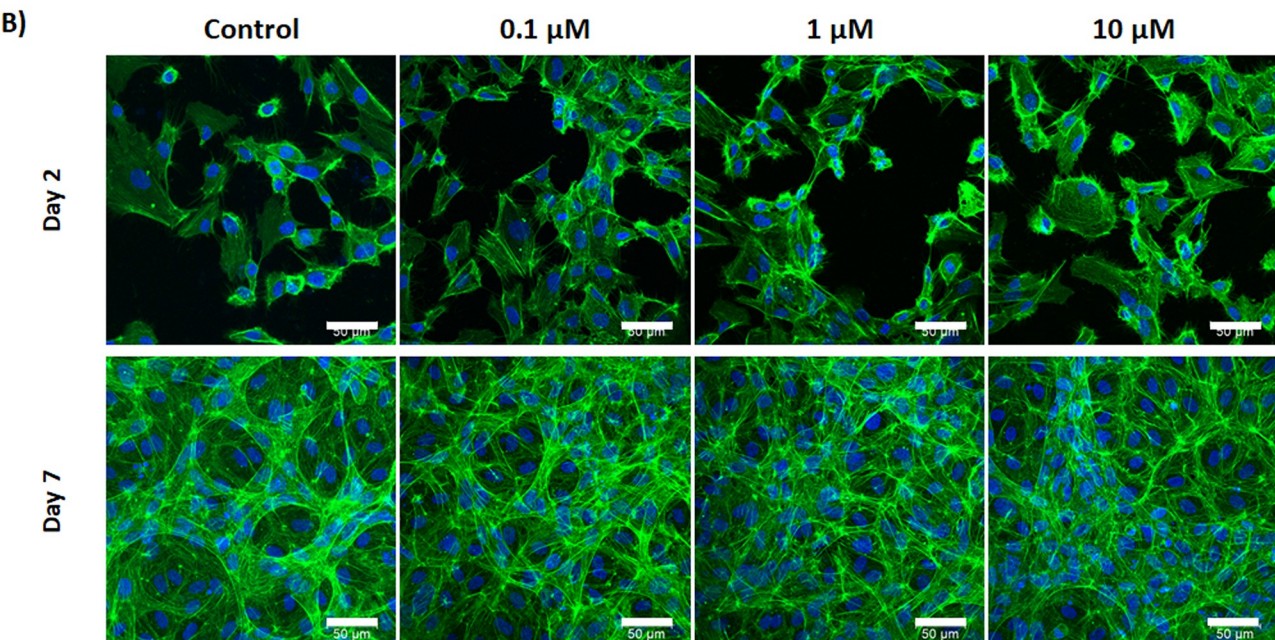

**Fig 1. HUVEC cultured with Cu²⁺ supplemented media.** (A) HUVEC viability cultured with different concentrations of $Cu^{2+}$ (0, 0.1, 1, 10, 100 and 200 μM) at day 2 and 7. (B) HUVEC morphology cultured with different non-toxic concentrations of $Cu^{2+}$ (0, 0.1, 1 and 10 μM) at day 2 and 7 (actin filament staining (green); nucleus (blue). Scale bar = 50 μm. Statistically significant differences were represented with * ($p < 0.05$).

for HUVEC at the end of culture (Fig 2A). Considering the lower viability of cells with BM, we decided to use GM for the subsequent assays.

With regards to HUVEC morphology, there were less number of cells with higher concentrations compared to control (Fig 2B). From day 2 to 7, HUVEC reached confluence and some cells proliferated over the monolayer with an elongated morphology although this was not observed with 50 μM.

## 3.2. Ion influence on angiogenesis gene expression

To further assess the potential of $Cu^{2+}$ and $Co^{2+}$ in stimulating angiogenic response in HUVEC, we analyzed the expression of three principal genes involved in the angiogenic response: HIF-1α, VEGF and PECAM-1. For this purpose, the gene expression profile was analyzed in a short period of hours (1, 2, 4, 10 and 24h), and subsequently, in longer periods (2 and 7 days), to assess the early and later angiogenic response, respectively.

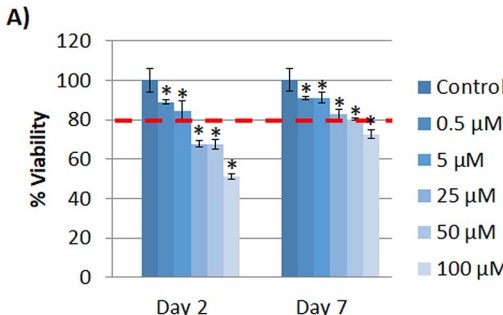

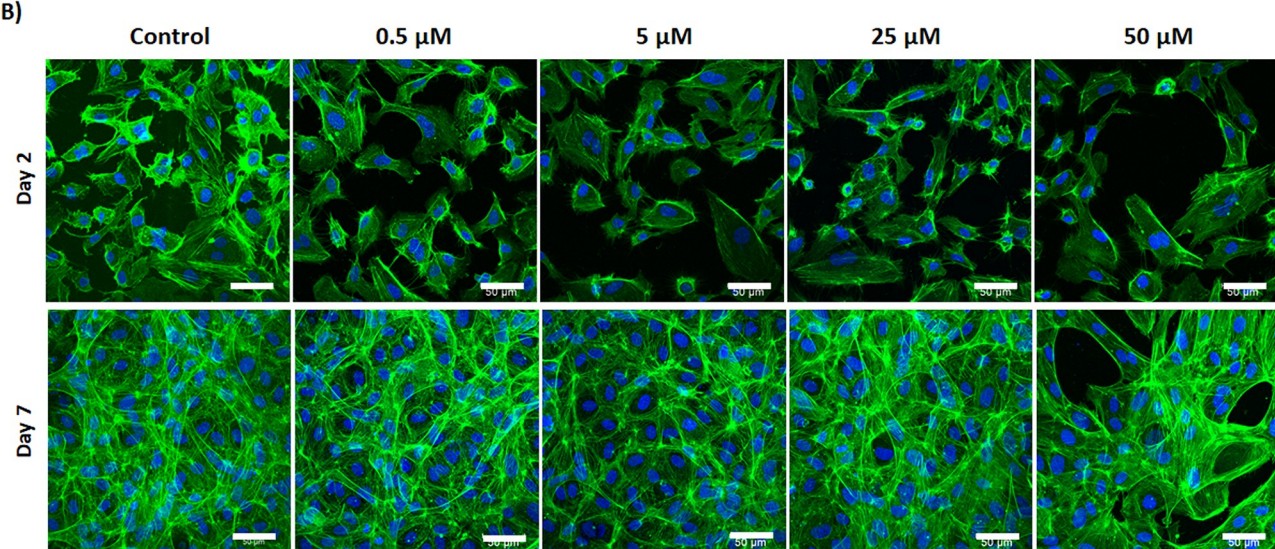

**Fig 2. HUVEC cultured with Co²⁺ supplemented media.** (A) HUVEC viability cultured with different concentrations of Co²⁺ (0, 0.5, 5, 25, 50 and 100 μM) at day 2 and 7 (n = 4). (B) HUVEC morphology cultured with different non-toxic concentrations of Co²⁺ (0, 0.5, 5, 25 and 50 μM) at day 2 and 7 (actin filament staining (green); nucleus (blue). Scale bar = 50 μm. Statistically significant differences were represented with * ($p < 0.05$).

**3.2.1. Copper (Cu²⁺).** In general, within the initial hours, HUVEC cultured with Cu²⁺ showed only a significant increase at 10h with 10 μM concentration (Fig 3A). Regarding the VEGF expression profile, it is worth mentioning that all Cu²⁺ concentrations induced a significantly higher VEGF expression within 1h in a dose-dependent manner compared to control, and the conditions of 0.1 and 10 μM could also increase it at 10h (Fig 3B). Finally, PECAM-1 expression was significantly up-regulated at different time points (1, 10 and 24h) with different conditions, being 10 μM able to increase it in all of them (Fig 3C).

At day 2, HUVEC HIF-1α expression analysis revealed similar expression with no significant differences compared to control (Fig 3D). At day 7, HIF-1α was significantly increased with the lowest concentration of 0.1 and 1 μM. Regarding VEGF expression at day 2, all Cu²⁺ concentrations significantly up-regulated it compared to control, in a dose-dependent manner (Fig 3E). Remarkably, 10 μM concentration induced 2.5 fold VEGF expression compared to control, and at the same time, it was higher than VEGF expression induced by 0.1 and 1 μM. Interestingly, at day 7 all the concentrations tested induced a higher VEGF expression compared to control. It is worth mentioning that 0.1 μM induced the highest effect with two-fold VEGF expression. Finally, PECAM-1 expression remained the same for all experimental conditions at day 2 and was significantly increased in a dose-dependent manner by 0.1 and 1 μM at day 7 (Fig 3F).

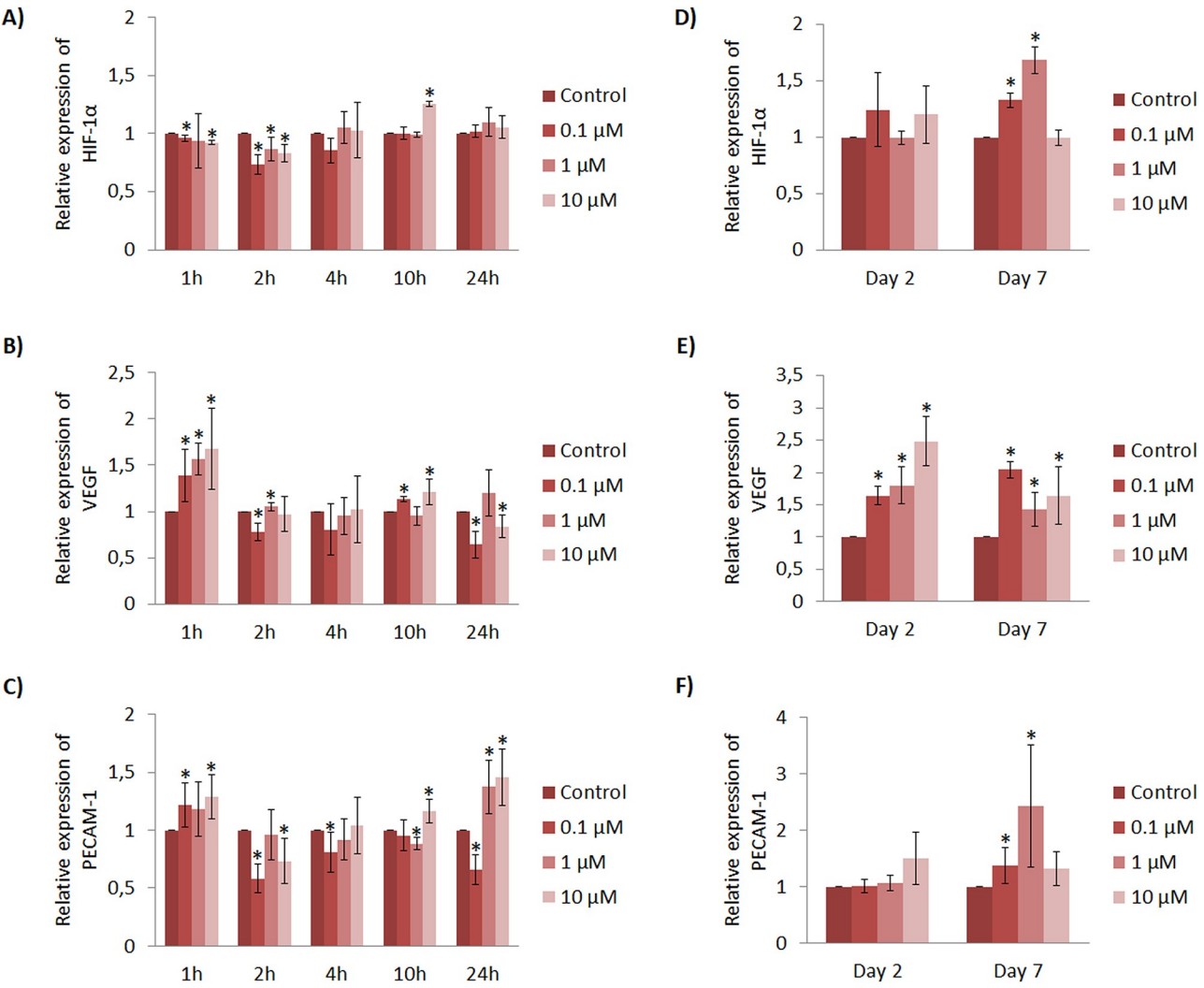

**Fig 3. HUVEC gene expression profile cultured with Cu²⁺ within short (hours) and longer (days) time points.** Different $Cu^{2+}$ concentrations were evaluated at 1, 2, 4, 10 and 24h for the gene expression of (A) HIF-1α, (B) VEGF and (C) PECAM-1. Different $Cu^{2+}$ concentrations were also evaluated at 2 and 7 days for the expression of (D) HIF-1α, (E) VEGF and (F) PECAM-1. Data represented correspond to fold-changes compared to each time point control. Statistically significant differences were represented with * ($p < 0.05$).

As HIF-1α is described to transcribe VEGF gene, we further assessed if there was a correlation between HIF-1α and VEGF expression (S1 Table). Although a significant correlation at 2 and 4h was found, there was no significant and strong correlation between the gene expression of HIF-1α and VEGF.

**3.2.2. Cobalt (Co²⁺).** When HUVEC were cultured with $Co^{2+}$ supplementation, results of HIF-1α expression within initial hours in general revealed an up-regulation at 1, 10 and 24h (Fig 4A). Regarding VEGF expression, there was an up-regulation at 1, 2 and 10h (Fig 4B). It is worth mentioning that, at 1h, all $Co^{2+}$ concentrations tested induced a higher VEGF expression compared to control, presenting 3 fold expression with 5 and 25 μM conditions. Finally, PECAM-1 expression was up-regulated at 1, 2, 10 and 24h, being worth to mention that 5 and 25 μM induced more than 3 fold expression at 1h (Fig 4C).

In later time points, at day 2, HUVEC HIF-1α expression analysis showed a gradual increase in a dose-dependent manner, being a significant up-regulation with 5, 25 and 50 μM

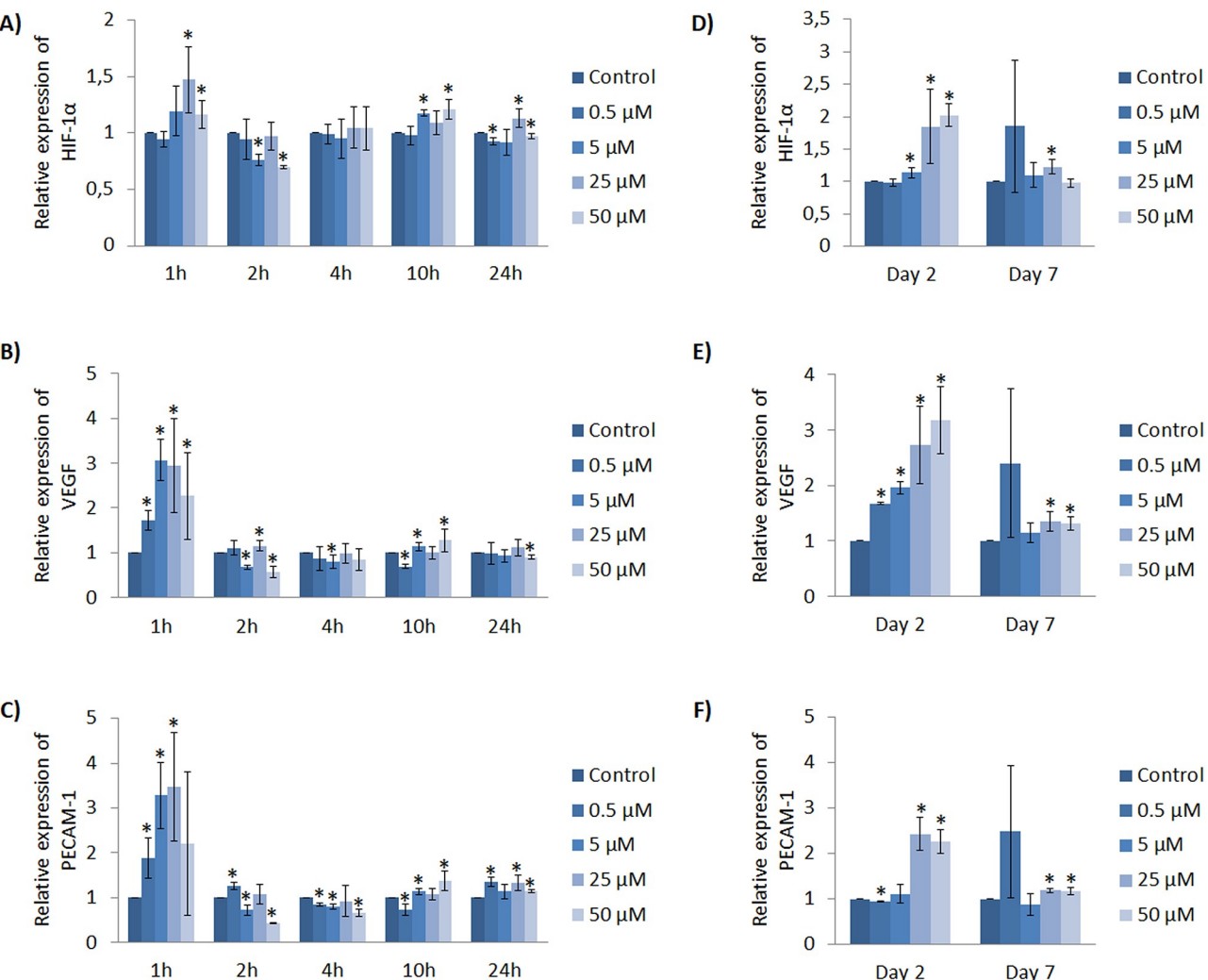

**Fig 4. HUVEC gene expression profile cultured with $Co^{2+}$ within short (hours) and longer (days) time points.** Different $Co^{2+}$ concentrations were evaluated at 1, 2, 4, 10 and 24h for the gene expression of (A) HIF-1α, (B) VEGF and (C) PECAM-1. Different $Co^{2+}$ concentrations were also evaluated at 2 and 7 days for the expression of (D) HIF-1α, (E) VEGF and (F) PECAM-1. Data represented correspond to fold-changes compared to each time point control. Statistically significant differences were represented with * ($p < 0.05$).

compared to control (Fig 4D). Later on, at day 7, only 25 μM induced a significant increase of HIF-1α expression. A similar trend was observed with VEGF expression. At day 2, there was a significant up regulation of VEGF for all concentrations tested in a dose-dependent manner, reaching 3-fold expression with 50 μM compared to its time point control (Fig 4E). At day 7, only 25 and 50 μM showed an up-regulation of VEGF expression. Finally, PECAM-1 expression at day 2 and 7 was significantly up-regulated by higher concentrations, specifically 25 and 50 μM, reaching almost 2.5 fold expression (Fig 4F).

As previously mentioned, HIF-1α is described to transcribe VEGF expression. For this reason, we further assessed if there was a correlation between HIF-1α and VEGF expression with $Co^{2+}$ supplementation (S2 Table). Statistical results show significant and strong correlation between the expressions of these two genes.

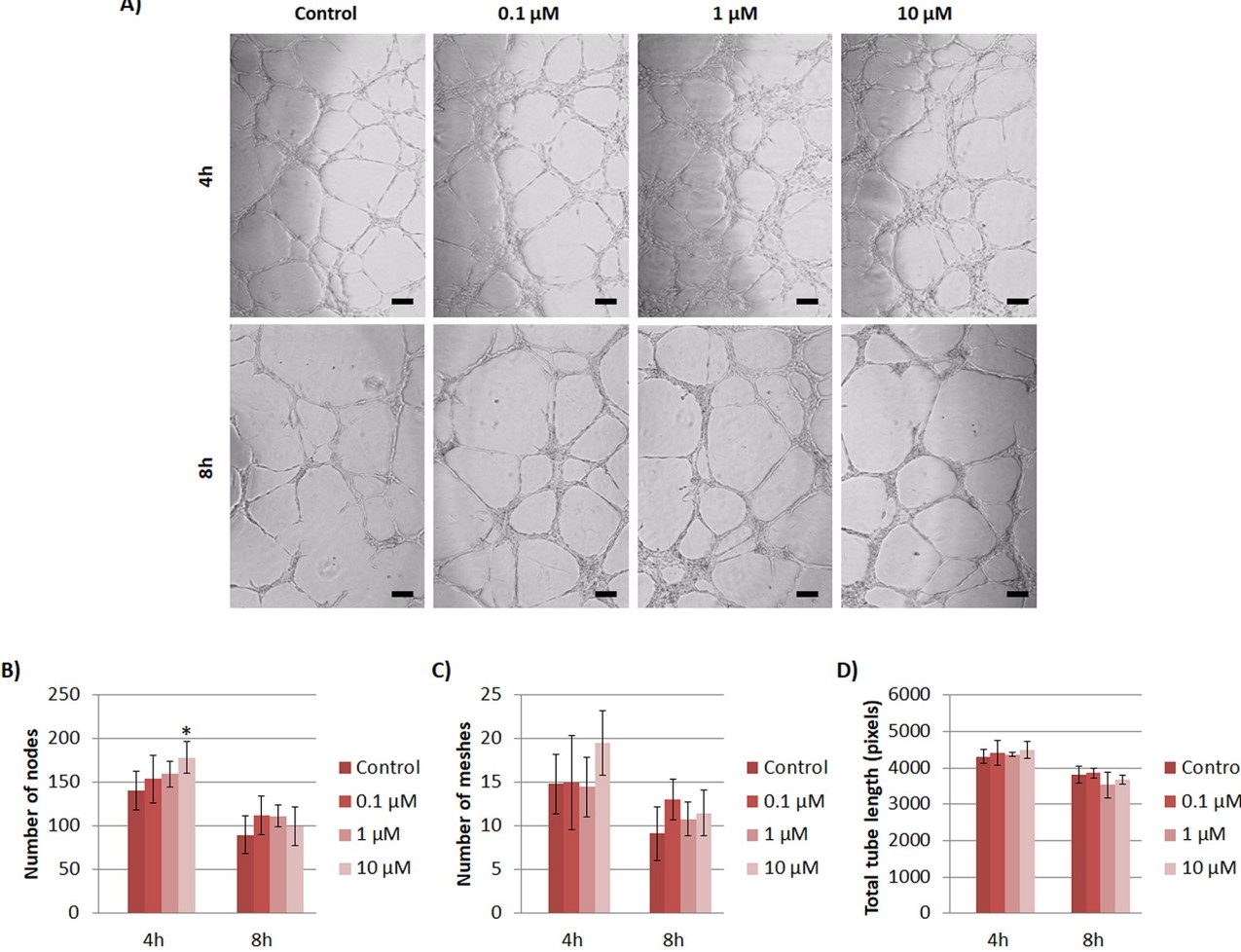

**Fig 5. Effects of $Cu^{2+}$ on tubule formation assay with HUVEC.** (A) Images of tubules formed under different non-toxic concentrations of $Cu^{2+}$ (0, 0.1, 1 and 10 μM) at 4 and 8h. (B) Analysis of number of nodes, (C) meshes and (D) total tube length of HUVEC under different $Cu^{2+}$ concentrations at 4 and 8 hours. Scale bar = 50 μm. Statistically significant differences were represented with * ($p < 0.05$).

### 3.3. Tubule formation ability

The ability of $Cu^{2+}$ and $Co^{2+}$ to promote formation of tubular-like structures was further assessed by culturing HUVEC on Matrigel substrate.

**3.3.1. Copper ($Cu^{2+}$).** In general, with $Cu^{2+}$ supplementation, thicker tubular structures were observed in comparison to control samples with phase contrast images at 4 and 8 hours (Fig 5A). The results of number of nodes at 4h resulted in an increase in a dose-dependent manner with $Cu^{2+}$ supplementation, although it was only statistically significant with 10 μM (Fig 5B). At 8h, the number of nodes decreased in all conditions compared to 4h. Regarding the number of meshes and total tube length, no significant differences were found between experimental conditions and control for the same time point (Fig 5C and 5D).

**3.3.2. Cobalt ($Co^{2+}$).** The assessment of $Co^{2+}$ influence in tubule-like structures formation showed, in general, broken vascular structures with 50 μM concentration at 4 and 8h (Fig 6A). Moreover, control at 8h also showed broken vascular network. Regarding the analysis of number of nodes, number of meshes and total tube length, a similar pattern was observed. More specifically, at 4h, there was a significant increase in the number of nodes with 0.5, 5 and

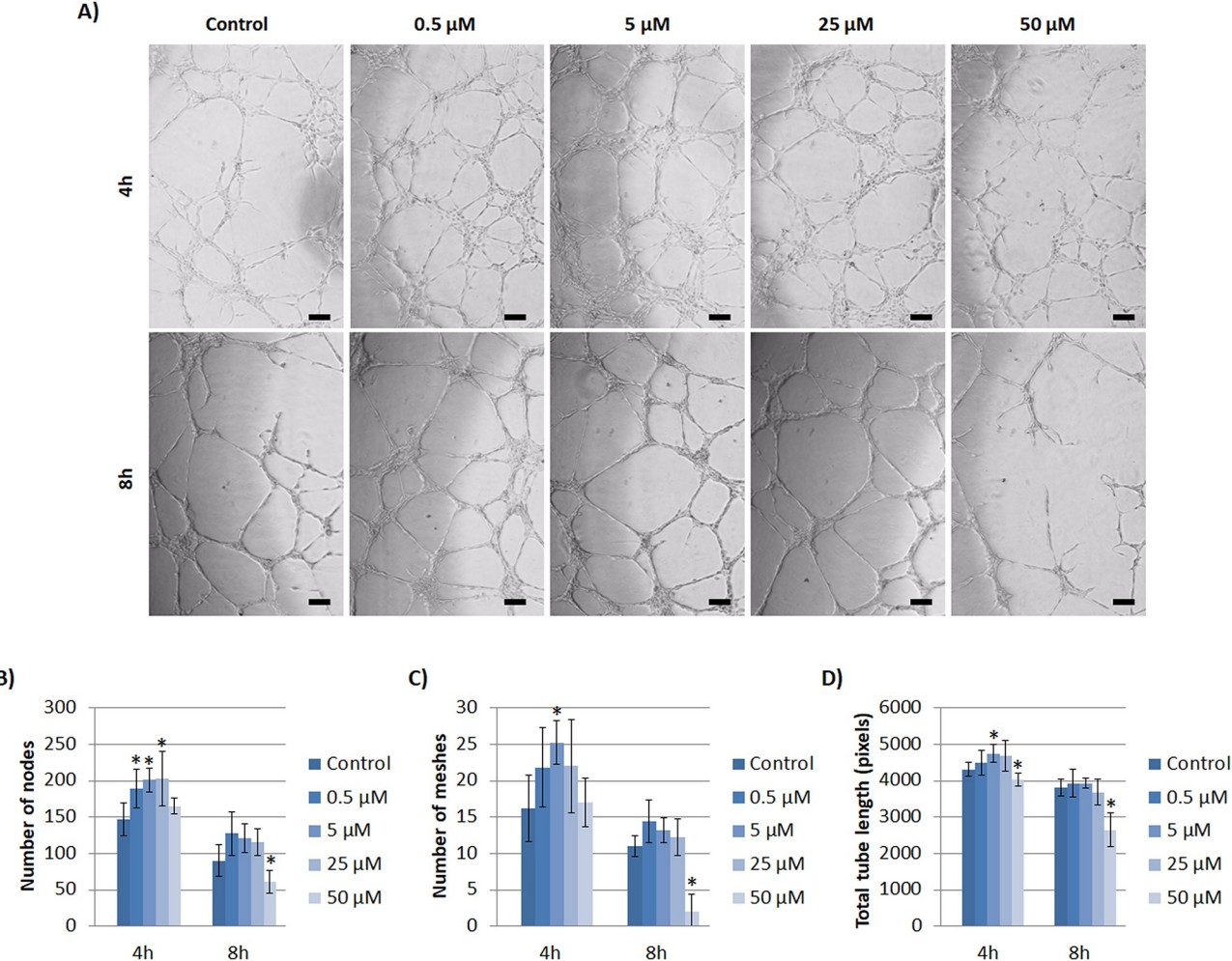

**Fig 6. Effects of Co$^{2+}$ on tubule formation assay with HUVEC.** (A) Images of tubules formed under different non-toxic concentrations of Co$^{2+}$ (0, 0.5, 5, 25 and 50 μM) at 4 and 8 hours. (B) Analysis of number of nodes, (C) meshes and (D) total tube length of HUVEC under different Co$^{2+}$ concentrations at 4 and 8 hours. Scale bar = 50 μm. Statistically significant differences were represented with * ($p < 0.05$).

25 μM (Fig 6B). A significant increase was also observed at 4h with 5 μM of Co$^{2+}$ regarding the number of meshes and total tube length (Fig 6C and 6D). At 8h, no significant increase was observed between experimental groups and control regarding the number of nodes, number of meshes and total tube length (Fig 6B–6D). However, a significant decrease was observed in all parameters with 50 μM.

### 3.4. Influence of dual ion culture on viability, proliferation and morphology

Additionally to the individual ion evaluation in angiogenesis response, we wanted to elucidate if the combination of both ions could have synergistic effects. To this purpose, we selected the concentration of 10 μM of Cu$^{2+}$, which demonstrated an important increase of VEGF at early time points, and the concentration of 25 μM of Co$^{2+}$, which also showed a significant improvement of VEGF and PECAM-1 at early and later time points. This combination was coded as Cu10/Co25. As we considered the possibility that the combination of these two higher

concentrations could be cytotoxic for cells due to their ability to form reactive oxygen species (ROS) when they are at high concentrations, we also assessed the combination of half of these concentrations, being 5 µM for $Cu^{2+}$ and 12.5 µM for $Co^{2+}$, coded as Cu5/Co12.5.

Results obtained with viability analysis showed that cells cultured with Cu5/Co12.5 and Cu10/Co25 reached a viability of almost 80% at day 2, but it was reduced down to 62% and 52% at day 7, respectively, being significantly lower compared to control (Fig 7A).

The influence of the combination of both ions on HUVEC morphology and organization was also evaluated with actin filament staining. At day 2, there were no apparent differences between experimental conditions (Fig 7B). At day 7, control samples reached confluence and HUVEC cells grew over it. Regarding both combination of ions, there were no differences regarding morphology compared to control.

### 3.5. Influence of dual ion culture on angiogenesis

To our surprise, the evaluation of angiogenic gene expression did not show any significant up or down-regulation of HIF-1α, VEGF or PECAM-1 for any of the combinations tested compared to control samples (Fig 8). As a final assessment, tubule formation capability of HUVEC under both ion combinations was also assessed with Matrigel as a substrate. Apparently, with phase contrast images there seemed to be no differences between all conditions tested (Fig 9A). Results of the analysis of number of nodes, number of meshes and total tube length resulted in non-statistically significant differences between experimental groups for each time point, confirming the previous observations (Fig 9A–9C).

### 4 Discussion

In order to use therapeutic ions for angiogenesis stimulation, a screening for non-toxic concentrations that at the same time induce a therapeutic effect is required. In this study, $Cu^{2+}$ and $Co^{2+}$ were selected, both described to be angiogenic, with the aim to elucidate how different concentrations affect the angiogenic response during time course (within hours and days), to determine which are the appropriate concentrations to be delivered when an ion delivering scaffold wants to be designed. As it is reported that the combination of some ions have a synergistic effects [20, 21], the combination of $Cu^{2+}$ and $Co^{2+}$ was also assessed in this study, as it has not been addressed before to the best of our knowledge.

The different assays were performed with growth media (GM) due to the presence of cell debris and detached cells with difficulties to proliferate with basal media (BM). Results showed non-toxic concentrations up to 10 µM for $Cu^{2+}$ and 50 µM for $Co^{2+}$ (**Figs** 1A and 2A). Decrease of cell viability with high concentrations of ions might be due to cellular toxicity induced by reactive oxygen species (ROS). In the presence of superoxide ($\cdot O_2-$) (produced by the mitochondria during cellular respiration) or reducing agents, such as glutathione (GSH) or ascorbic acid (generally found in cells [22] and in cell culture media that we used for the experiments, respectively), $Cu^{2+}$ can be reduced to $Cu^+$, which in turn, can catalyze the formation of hydroxyl radicals (OH·) from hydrogen peroxide ($H_2O_2$) (produced during normal cell metabolism [23]), through Haber-Weiss reaction [24]:

$$\cdot O_{2-} + Cu^{2+} \rightarrow O_2 + Cu^+$$

$$Cu^+ + H_2O_2 \rightarrow Cu^{2+} + OH^- + OH^.$$

Similarly, $Co^{2+}$ has also been described to be involved in ROS formation [25, 26]. More specifically, with the presence of oxygen, $Co^{2+}$ might generate the radical intermediate $Co^+$-OO·

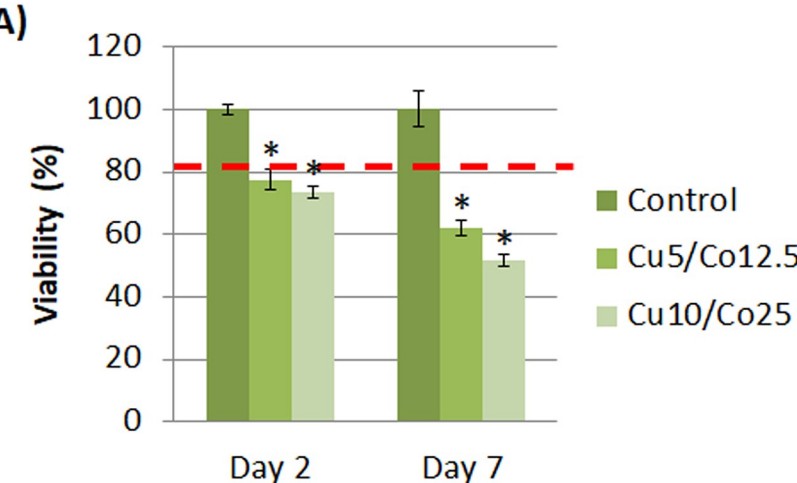

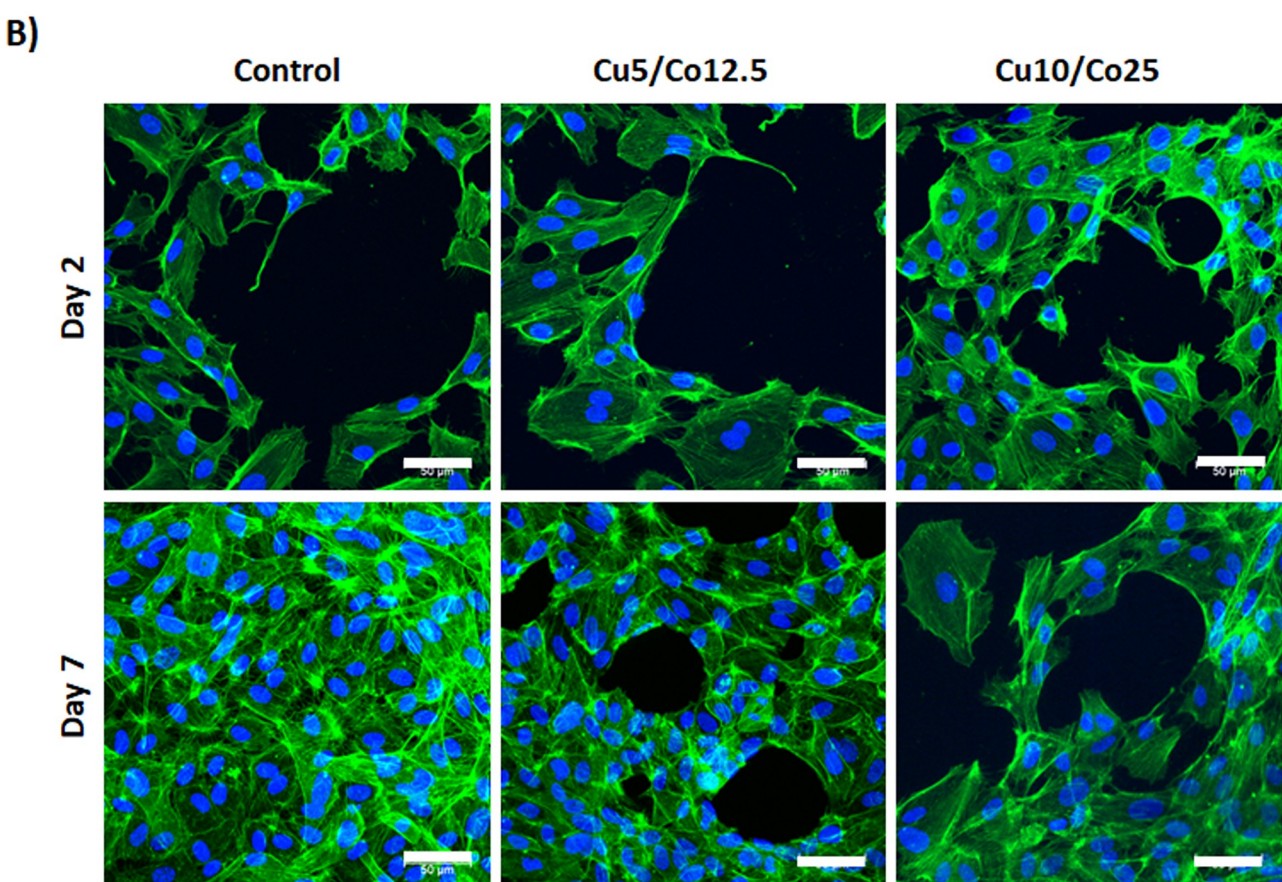

**Fig 7. HUVEC viability, cell number and morphology with Cu²⁺ and Co²⁺ supplementation.** (A) Viability and (B) morphology of HUVEC under different combinations of Cu²⁺ and Co²⁺ (actin filament staining (green); nucleus (blue). Scale bar = 50 μm. Abbreviations: Cu5/Co12.5 = 5 μM of Cu²⁺ and 12.5 μM of Co²⁺; Cu10/Co25 = 10 μM of Cu²⁺ and 25 μM of Co²⁺. Statistically significant differences were represented with * (p<0.05).

species [27]:

$$Co^{2+} + O_2 \rightarrow Co^+ + O_2^. \rightarrow Co^+ - OO^.$$

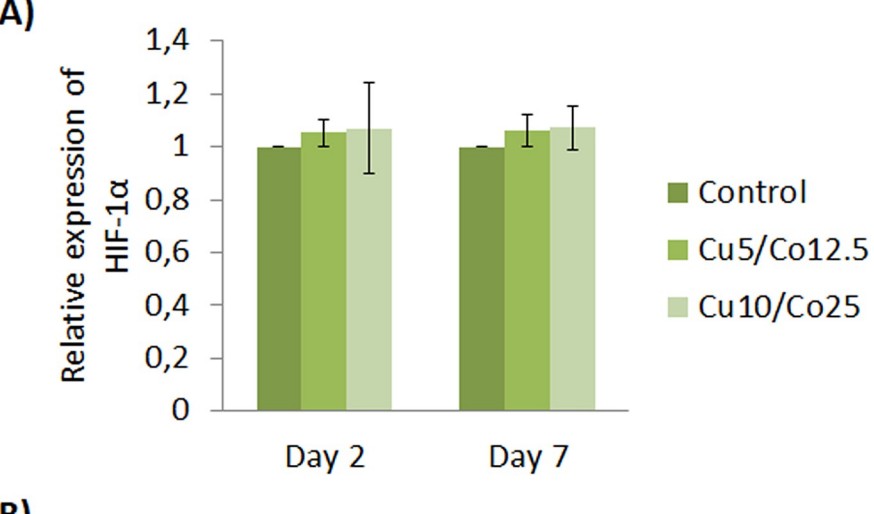

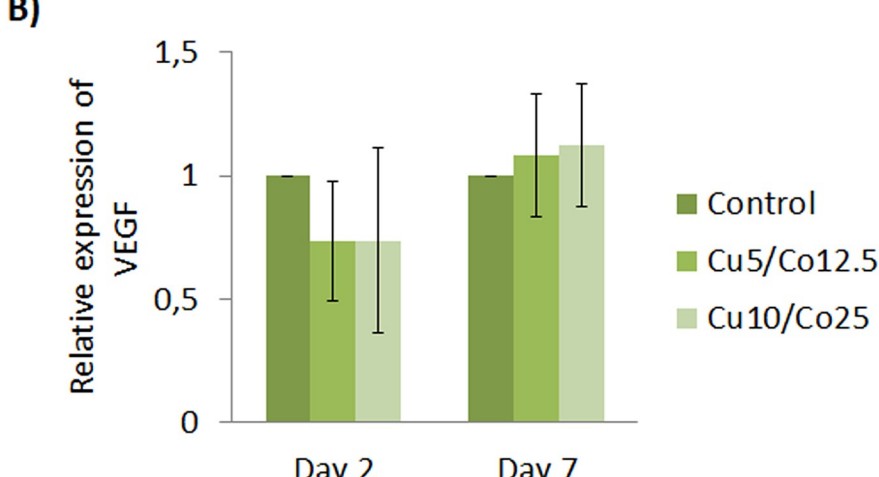

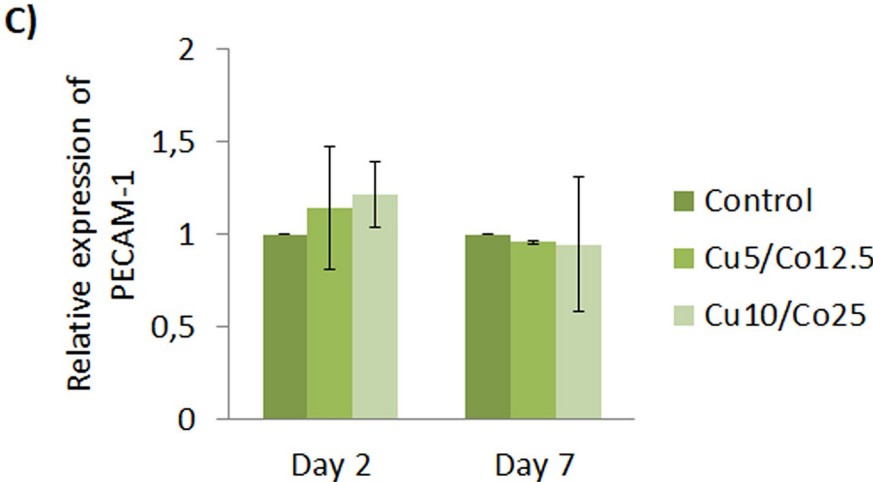

**Fig 8. HUVEC gene expression.** Different $Cu^{2+}$ and $Co^{2+}$ concentrations were evaluated at 2 and 7 days for the expression of (A) HIF-1α, (B) VEGF and (C) PECAM-1. Data represented correspond to fold-changes compared each time point control. Abbreviations: Cu5/Co12.5 = 5 μM of $Cu^{2+}$ and 12.5 μM of $Co^{2+}$; Cu10/Co25 = 10 μM of $Cu^{2+}$ and 25 μM of $Co^{2+}$.

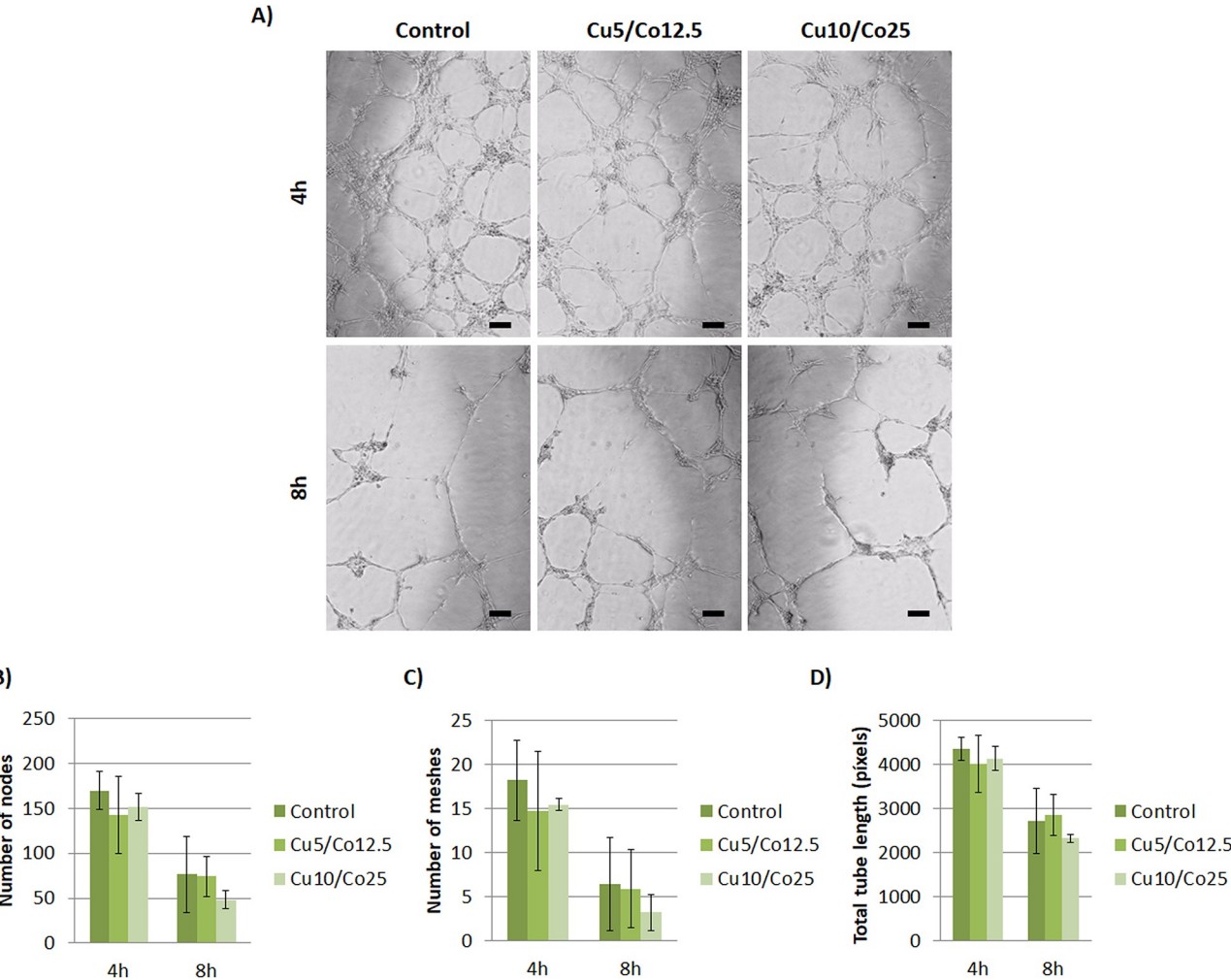

**Fig 9. Effects of Cu²⁺ and Co²⁺ on tubule formation assay with HUVEC.** (A) Images of tubules formed under different concentrations of $Cu^{2+}$ and $Co^{2+}$ at 4 and 8 hours. (B) Analysis of number of nodes, (C) meshes and (D) total tube length of HUVEC at 4 and 8 hours. Abbreviations: Cu5/Co12.5 = 5 μM of $Cu^{2+}$ and 12.5 μM of $Co^{2+}$; Cu10/Co25 = 10 μM of $Cu^{2+}$ and 25 μM of $Co^{2+}$. Scale bar = 50 μm.

With the presence of superoxide dismutase enzyme (SOD) (an antioxidant enzyme present in cells [28]), it can catalyze the decomposition of $Co^+$-$OO^·$ species to $H_2O_2$ and $Co^+$:

$$Co^+ - OO^· \rightarrow H_2O_2 + Co^+$$

Finally, it was proposed that $Co^+$ could participate in the formation of $OH^·$ radicals through Fenton reaction:

$$Co^+ + H_2O_2 \rightarrow Co^{2+} + OH^· + OH^-$$

Results of HUVEC viability when both ions were combined support the proposed equations of ROS generation by $Cu^{2+}$ or $Co^{2+}$ discussed in the literature. Surprisingly, HUVEC viability was considerably reduced in the presence of both ions (Fig 7A), compared to the viability with individual ions (**Figs** 1A and 2A). As observed in ROS generation equations, $Cu^{2+}$ generates $O_2$, whereas $Co^{2+}$ consumes it. Therefore, there is the possibility that the presence of both ions might feedback the formation of ROS. Furthermore, it is widely reported that the generation

of ROS can result in DNA damage, lipid peroxidation, depletion of protein sulfhydryls and other effects that leads to apoptosis of cells [27], and that might explain the decreased viability of cells when the concentrations of ions were increased. To confirm low viability due to ROS formation, further studies including quantification of ROS species will need to be undertaken.

Comparing viability results of HUVEC cultured with $Cu^{2+}$ with published ones, we found similarities with another study. *Similarly*, *they* reported a decreased HUVEC viability when cultured with concentrations equivalent to 7, 21 and 63 μM compared to control [29]. However, our results differ from other studies reporting viability of HUVEC up to 50 μM [30] or 500 μM [31]. These differences might be due to the time of exposure of $Cu^{2+}$, which was relatively short for the later studies (48h), whereas in our study and the one with similar results [29] the $Cu^{2+}$ exposure was up to 7 and 14 days respectively. Regarding $Co^{2+}$, similar to our results, Peters *et al.*, described a proliferation decrease in a dose-concentration manner with concentrations of 10, 50, 100, 300 and 700 μM [32]. Conversely, Tao Zan *et al* reported an enhanced proliferation of EC in a dose-dependent manner with $Co^{2+}$ concentrations of 50, 100 and 200 μM [33]. These differences might be due to the degree of differentiation of EC. Endothelial progenitor cells (EPC) are described to possess higher antioxidant capacity than mature endothelial cells such as HUVEC [34], therefore, being resistant to oxidant cytotoxicity generated by ROS. Tao Zan *et al* used EPC for their study, and that might explain the absence of toxicity up to 200 μM.

Non-toxic concentrations of $Cu^{2+}$ (≤10 μM) demonstrated an enhancement of angiogenic response both in a gene expression level and in the ability to form tubular structures. Regarding the gene expression our results demonstrated that, in a wide overview, higher concentrations (10 μM) of $Cu^{2+}$ have a significant contribution in angiogenesis at early time points observed with VEGF and PECAM-1 expression (up to day 2), whereas lower concentrations (0.1 and 1μM) have a more therapeutic effect at later stages (day 7) (Fig 3). Statistical analysis revealed no correlation between HIF-1α and VEGF expression (S1 Table), however, a previous study described that $Cu^{2+}$ can enhance angiogenic response by the stabilization and increase of HIF-1α, which in turn triggers VEGF transcription [8]. Our findings are in agreement with a later published study, in which authors suggested that $Cu^{2+}$ might be required for the binding of HIF-1α to hypoxia response element (HRE) sequence of target genes such as VEGF, but its deprivation did not affect the expression or stability of HIF-1α (total amount of HIF-1α in cells) but it did decrease VEGF expression [35]. Therefore, authors suggest that $Cu^{2+}$ does not increase HIF-1α expression, but it is required for HIF-1 transcriptional complex formation. This might explain why we could observe an increase of VEGF although it was not correlated with HIF-1α expression. In addition, we observed a significant increase of PECAM-1 expression with higher doses at early times (10 μM at 24h) and with lower doses at later times (0.1 and 1 μM at day 7). It has been described that PECAM-1 is involved in endothelial cell migration and angiogenesis [36–38], and a reduction of blood vessel formation was reported when being inhibited [39]. Therefore, PECAM-1 expression results further suggests an enhancement of angiogenic response with $Cu^{2+}$. Gene expression results of VEGF and PECAM-1 seem to be in accordance with results obtained with tubule formation assay (Fig 5). More specifically, an increase of number of nodes and meshes was observed with the high concentration of 10 μM at 4h, whereas at later time point, the increase was observed with 0.1 μM. Therefore, our results suggest that $Cu^{2+}$ is able to establish a higher degree of vascular network connections with concentrations ranging from 0.1 to 10 μM.

Non-toxic concentrations of $Co^{2+}$ (≤50 μM) could also demonstrate an enhancement of angiogenic response, although differences were found with the highest therapeutic dose between gene expression results and the ability to form tubular structures. Regarding results of gene expression induced by $Co^{2+}$, although lower concentrations can enhance the angiogenic

response within hours, it seems that higher concentrations of 25 and 50 μM enhance the response either at 2 and 7 days through the increase of HIF-1α, VEGF and PECAM-1 expression (Fig 4). Remarkably, we found a strong and significant correlation between HIF-1α and VEGF expression (S2 Table). These results are similar to a previous study demonstrating that $Co^{2+}$ induced increased HIF-1α and VEGF expression in a dose-dependent manner, although the concentrations that they used were higher, specifically 50, 100 and 200 μM [33]. Authors mentioned that $Co^{2+}$ act as prolyl hydroxylation inhibitor, hence inhibiting HIF-1α degradation and stabilizing it, with subsequent VEGF expression. Moreover, the up-regulation of PECAM-1 expression with high concentrations further indicates an enhancement of angiogenic response. Alternatively, the results obtained from tubule formation assay indicated that $Co^{2+}$ have an effect in vascular structure formation (Fig 6). More specifically, there was an increase of number of nodes, number of meshes and total tube length with 0.5, 5 and 25 μM at early time of 4h. Importantly, the highest concentration of 50 μM induced an impairment of angiogenesis at 8h. This impairment was in accordance with a previous study where the in-vitro capillary formation of human EC was impaired with $Co^{2+}$ concentrations of 50, 100 and 300 μM [32]. The authors hypothesized that $Co^{2+}$ exposure could induce interference of integrin signaling, which is normally regulated by divalent metal ions such as $Ca^{2+}$, $Mg^{2+}$ and $Mn^{2+}$. Integrins recognize and respond to a variety of extracellular matrix proteins [40] and their signaling interference can lead to a loss of EC adhesion [41], which might be an explanation for the impairment of vascular network structures formation with the highest concentration of $Co^{2+}$.

Finally, as we observed an enhanced angiogenic response with individual culture of $Cu^{2+}$ and $Co^{2+}$ with HUVEC, we further evaluated the response when combined together. As far as we know, no previous studies are reported analyzing the role of these two ions combined together, neither as cell culture media supplement or as delivered from scaffolds. To our surprise, we could not observe any significant up or down-regulation of HIF-1α, VEGF or PECAM-1 between conditions during all time course of the experiment (Fig 8), or any significant difference of number of nodes, number of meshes and total tube length with the *in vitro* tubule formation assay compared to control (Fig 9). As we observed differences when both ions were cultured individually, these results point to the likelihood that both ions are mutually inhibiting or interacting with each other when being together. A possible hypothesis is that copper and cobalt might interact with some components of the cell culture media forming new complexes. In this regard, it is described in the literature that copper and cobalt can interact with each other with the presence of histidine, forming a new complex Co-Cu-His [42]. As aminoacids, being histidine one of them, are added in the cell culture media formulations as essential nutrients [43], the possibility of forming the complex Co-Cu-His is possible. This would eventually explain why there is no increase in the angiogenesis response, as copper and cobalt might not be found freely to be able to enhance the HIF-1α binding to HRE (copper) and to inhibit the HIF-1α degradation (cobalt). However, future research is needed to test this hypothesis regarding the lack of synergy when both ions are combined together.

The results obtained in this study regarding the individual effects of ions provide useful information for the design of scaffolds for the delivery of $Cu^{2+}$ or $Co^{2+}$ in tissue engineering field. Fig 10 shows schematic representation of the different doses of $Cu^{2+}$ and $Co^{2+}$ that proved to enhance the angiogenic response, either increasing VEGF or PECAM-1 and improving the ability to form tubule structures, through time course up to 7 days. The released amount from scaffolds within these ranges of concentrations might have the potential to stimulate blood vessel formation.

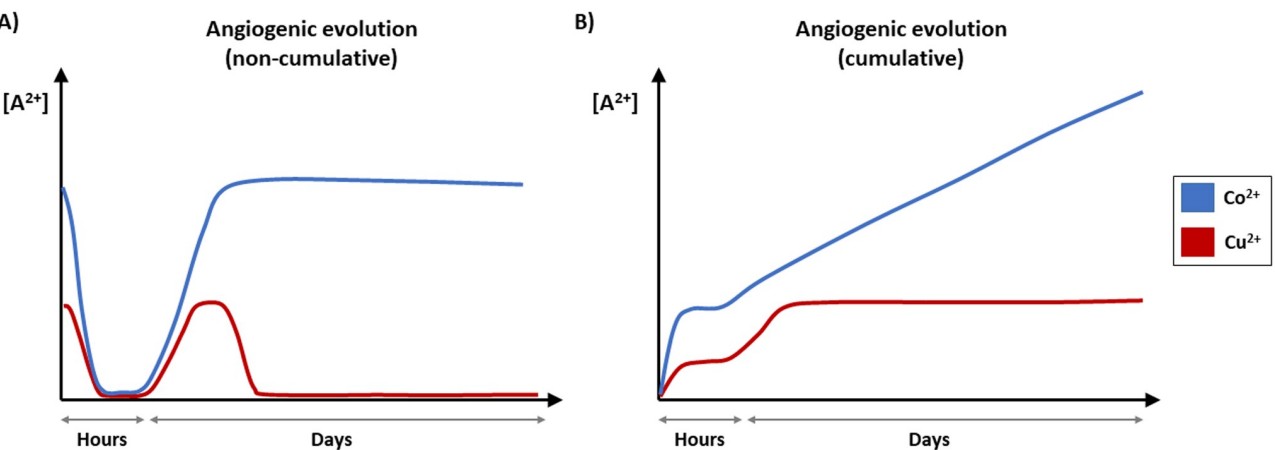

**Fig 10. Schematic representation of angiogenic therapeutic doses of Cu²⁺ and Co²⁺.** (A) Profile of doses of $Cu^{2+}$ and $Co^{2+}$ that enhanced the angiogenic response within hours and days and (B) the corresponding cumulative release. Representation made based on VEGF and PECAM-1 gene expression and tubule formation assay results.

## 5 Conclusions

The results of this study indicate that $Cu^{2+}$ and $Co^{2+}$ ions individually enhanced the angiogenic response in HUVEC cells with different concentrations, although results showed that some concentrations have earlier therapeutic effects than others. Higher concentrations of $Cu^{2+}$ have an early angiogenic effect, whereas lower concentrations are required for the increase of angiogenic response at later time points. Alternatively, in a wide overview, higher doses of $Co^{2+}$ stimulates an angiogenic response at early and later times. These results have important implications regarding the design of $Cu^{2+}$ or $Co^{2+}$ delivery scaffolds when blood vessel formation wants to be stimulated at early or later times. Otherwise, our results indicate that the combination of both ions should be avoided.

## Supporting information

**S1 Fig. Optical images of HUVEC cultured with Cu²⁺ supplemented basal media.**
(TIFF)

**S2 Fig. Optical images of HUVEC cultured with Cu²⁺ supplemented growth media.**
(TIFF)

**S3 Fig. HUVEC cultured with Cu²⁺ with basal medium.** Viability of HUVEC cultured with basal medium supplemented with $Cu^{2+}$. Statistically significant differences were represented with * ($p < 0.05$).
(TIFF)

**S4 Fig. Optical images of HUVEC cultured with Co²⁺ supplemented basal media.**
(TIFF)

**S5 Fig. Optical images of HUVEC cultured with Co²⁺ supplemented growth media.**
(TIFF)

**S6 Fig. HUVEC cultured with Co²⁺ with basal medium.** Viability of HUVEC cultured with basal medium supplemented with $Co^{2+}$. Statistically significant differences were represented with * ($p < 0.05$).
(TIF)

**S1 Table. HIF-1α and VEGF expression correlation results with Cu²⁺ supplementation.**
Symbols: ns = non-significant (p>0.05); * = p<0.05; ** = p<0.01.
(TIF)

**S2 Table. HIF-1α and VEGF expression correlation results with Co²⁺ supplementation.**
Symbols: ns = non-significant (p>0.05); * = p<0.05; ** = p<0.01.
(TIF)

## Author Contributions

**Conceptualization:** Roman A. Perez.

**Funding acquisition:** Roman A. Perez.

**Investigation:** Roman A. Perez.

**Methodology:** Elia Bosch-Rué, Begoña María Bosch-Canals.

**Project administration:** Elia Bosch-Rué, Roman A. Perez.

**Resources:** Roman A. Perez.

**Validation:** Leire Díez-Tercero, Raquel Rodríguez-González, Begoña María Bosch-Canals, Roman A. Perez.

**Writing – original draft:** Elia Bosch-Rué, Leire Díez-Tercero, Raquel Rodríguez-González, Begoña María Bosch-Canals.

**Writing – review & editing:** Leire Díez-Tercero, Raquel Rodríguez-González, Begoña María Bosch-Canals, Roman A. Perez.

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
