## [Decision Letter · Decision Letter 0]

8 Sep 2021

PONE-D-21-25451Assessing the potential role of copper and cobalt in stimulating angiogenesis for tissue regenerationPLOS ONE

Dear Dr. Perez,

Thank you for submitting your manuscript to PLOS ONE. After careful consideration, we feel that it has merit but does not fully meet PLOS ONE’s publication criteria as it currently stands. Therefore, we invite you to submit a revised version of the manuscript that addresses the points raised during the review process.

We look forward to receiving your revised manuscript.

Kind regards,

Selvaraj Vimalraj

Academic Editor

PLOS ONE

Journal Requirements:

2.In your Data Availability statement, you have not specified where the minimal data set underlying the results described in your manuscript can be found. PLOS defines a study's minimal data set as the underlying data used to reach the conclusions drawn in the manuscript and any additional data required to replicate the reported study findings in their entirety. All PLOS journals require that the minimal data set be made fully available. For more information about our data policy, please see http://journals.plos.org/plosone/s/data-availability.

Reviewers' comments:

Reviewer's Responses to Questions

**Comments to the Author**

1. Is the manuscript technically sound, and do the data support the conclusions?

Reviewer #1: Yes

Reviewer #2: Yes

2. Has the statistical analysis been performed appropriately and rigorously? 

Reviewer #1: Yes

Reviewer #2: Yes

3. Have the authors made all data underlying the findings in their manuscript fully available?

Reviewer #1: Yes

Reviewer #2: Yes

4. Is the manuscript presented in an intelligible fashion and written in standard English?

Reviewer #1: Yes

Reviewer #2: Yes

5. Review Comments to the Author

Reviewer #1: The manuscript titled on "Assessing the potential role of copper and cobalt in stimulating angiogenesis for tissue

regeneration" is an interesting article. However there are few questions should be addressed by author.

1) What is the rationale for choosing copper and cobalt?. Its already available for treating several diseases.

2) Why an author chosen HUVEC cell line, rather than other cell lines?

3) Do an author checked the protein expression of vegf after stimulation in endothelial cells ?

Reviewer #2: MAJOR COMMENTS

This study by Bosch-Rué et al. examined the role of copper (Cu2+) and cobalt (Co2+) for inducing angiogenesis process and tested their hypothesis in vitro using Human Umbelical vein endothelial cell (HUBEC) by gene expression analysis and the assessment of tubular formation structures. The findings could be interesting and important for understanding the molecular basis of biological active metal ions like copper (Cu2+) and cobalt (Co2+) in angiogenesis and subsequent future therapeutic application in tissue regeneration. However, the data represented in this manuscript is only based on in vitro study and assessment of individualistic and synergistic effect of copper (Cu2+) and cobalt (Co2+) in angiogenesis. It could have been better if the in vitro tested dosage of copper (Cu2+) and cobalt (Co2+) is tested in a tissue regeneration model using synthetic grafts.

Although the data related separate dosage effect of copper (Cu2+) and cobalt (Co2+) in angiogenesis would be useful for the scientific community but synergistic effect did not show any significant improvement. Thus, a mechanistic data of the insignificant combinatorial role of copper (Cu2+) and cobalt (Co2+) in angiogenesis could have been interesting and that could add the missing novel insights in this existing story.

MINOR COMMENTS

1. All fluorescence images should be marked as stained accordingly. Like actin filament (Green) / phalloidin (Green) and DAPI (Blue).

2. Page 21, Line 432 & 433 “Stähli et al. reported a decreased HUVEC viability when 433 cultured with concentrations equivalent to 7, 21 and 63 μM compared to control (29).” Formatting should be corrected as per journal requirement.

6. PLOS authors have the option to publish the peer review history of their article (what does this mean?). If published, this will include your full peer review and any attached files.

Reviewer #1: No

Reviewer #2: **Yes: **Subhra Prakash Hui

---

## [Author Response · Author response to Decision Letter 0]

22 Sep 2021

Reviewer #1

The manuscript titled on "Assessing the potential role of copper and cobalt in stimulating angiogenesis for tissue regeneration" is an interesting article. However there are few questions should be addressed by author.

1) What is the rationale for choosing copper and cobalt?. Its already available for treating several diseases.

We would like to thank the reviewer for revising the manuscript and for the valuable feedback. The reason for choosing copper and cobalt is because it is well established that these ions are able to induce hypoxic conditions and subsequently an angiogenesis response. However, although there are several studies that have incorporated cobalt or copper into their scaffolds, the angiogenic response and or release has not been measured during the entire time course. Instead, they are generally measured at the end of the experiment or in a single time point during the ion exposure. These studies do not provide enough data for new scaffold design in terms of delivering appropriate doses of copper or cobalt at specific times that are non-toxic and promotes the angiogenic response at the same time. Therefore, we consider interesting and useful to study the angiogenic response during the first period of time (hours) and later time points (days). Special importance is the doses released during the first hours, as burst release can happen from scaffolds and induce toxicity. Hence, establishing a therapeutic non-toxic range is necessary. We mentioned these characteristics as relevant in the introduction (line 61-66):

“Despite the significant number of scaffolds that have incorporated these ions into their structure, the optimum concentrations required at each time point of the angiogenic response are still unknown. Generally, ion release from scaffolds is measured as a cumulative release after several days, obtaining the angiogenic response at the end point of the experiment (11,12) or in a single time point during the course of the experiment (13–16), which is generally never in the initial hours (17–19)”.

A part from that, there is no previous report combining both ions, copper and cobalt, together and assessing if there exist a synergistic effect, previous demonstrated to be an alternative strategy of angiogenesis stimulation with the combination of other ions. As it has been demonstrated that both ions have an angiogenic potential when individually used, we wanted to elucidate if their combination could enhance the angiogenic response or not. This information is also useful for a new ion-doped scaffold design, in order to combine or avoid both ions together. A new nuance has been added regarding the combination of the two ions (line 85).

“Furthermore, the possible synergistic combinatorial effect of these non-toxic ion concentrations was assessed, in order to open the possibility of combining ions to stimulate angiogenesis as this has not been studied before”.

2) Why an author chosen HUVEC cell line, rather than other cell lines?

In order to study the angiogenic response due to ion stimuli, an endothelial cell line was needed. In this regard, HUVEC cells have been extensively used, with approximately 5432 search results in Pubmed for “HUVEC and Angiogenesis”:

https://pubmed.ncbi.nlm.nih.gov/?term=%28HUVEC%29+and+%28angiogenesis%29&sort=pubdate

Moreover, in a recent review (Medina Leyte DJ, Domínguez Pérez M, Mercado I, Villarrreal Molina MT, Jacobo Albavera L. Use of Human Umbilical Vein Endothelial Cells (HUVEC) as a Model to Study Cardiovascular Disease: a review. Appl Sci. 2020;10(3)) is mentioned that HUVEC faithfully represents human endothelial cell behavior as compared to other cell lines, and that they are useful for angiogenesis model, particularly suitable for in vitro screening, the evaluation of ultrastructure of capillary formation, amongst others.

For these reasons, we decided to use HUVEC cell line as it is considered a good in vitro model for angiogenesis studies, with similar behavior of other endothelial cell lines.

3) Do an author checked the protein expression of vegf after stimulation in endothelial cells?

We did not check the protein expression of VEGF after ion stimulation. We consider that protein expression of vegf is to verify the functionality of the cells, providing information not only if the genes are expressed but also if cells will behave induce the production of a blood vessel. Alternatively, we decided to test functionality by performing the tubule formation assay which is mainly mediated by VEGF. Moreover, we used a growth factor reduced matrigel in order to observe clearly the effects of individual ions in vascular-like network formation. Therefore, with tubule formation results we have evidence that the angiogenic response is enhanced, especially with cobalt ion.

Reviewer #2

MAJOR COMMENTS

This study by Bosch-Rué et al. examined the role of copper (Cu2+) and cobalt (Co2+) for inducing angiogenesis process and tested their hypothesis in vitro using Human Umbelical vein endothelial cell (HUBEC) by gene expression analysis and the assessment of tubular formation structures. The findings could be interesting and important for understanding the molecular basis of biological active metal ions like copper (Cu2+) and cobalt (Co2+) in angiogenesis and subsequent future therapeutic application in tissue regeneration. However, the data represented in this manuscript is only based on in vitro study and assessment of individualistic and synergistic effect of copper (Cu2+) and cobalt (Co2+) in angiogenesis. It could have been better if the in vitro tested dosage of copper (Cu2+) and cobalt (Co2+) is tested in a tissue regeneration model using synthetic grafts.

Although the data related separate dosage effect of copper (Cu2+) and cobalt (Co2+) in angiogenesis would be useful for the scientific community but synergistic effect did not show any significant improvement. Thus, a mechanistic data of the insignificant combinatorial role of copper (Cu2+) and cobalt (Co2+) in angiogenesis could have been interesting and that could add the missing novel insights in this existing story.

We would like to thank the reviewer for revising the manuscript and for the valuable feedback given.

We agree with the reviewer that the tested dosage of copper and cobalt would have been better by using synthetic grafts. However, to this purpose, a previous knowledge about the correct doses that need to be delivered over time is necessary to design a proper scaffold that enhances the angiogenic response. As previously mentioned and referenced in the manuscript (lines 63-66), studies using copper or cobalt ions in their scaffolds do not provide enough data for that, usually reporting the doses delivered and angiogenesis response at the end of the experiment as a cumulative release or in a single time point. Therefore, we considered that providing this data over short period of time (hours) and longer time points (days) is crucial when a scaffold to deliver these ions for angiogenesis purposes wants to be developed. So from our results, synthetic grafts delivering those ions can be more properly designed.

Regarding the lack of synergy with the combination of both ions, we agree with the reviewer that we did not add a possible explanation of why that might occur and we added new information regarding this. We found that both copper and cobalt can interact with each other with the presence of some α-aminoacids, specifically α-histidine (see reference below).

Lukasiewicz A, Wozniak I. Interaction between the metals in cobalt(II)-copper(II), cobalt(II)-chromium(III) and iron(II)-chromium(III) cation pairs in the presence of α-amino acids and acetat anion in queous solutions. J Chem Soc chemmical Commun. 1989;(5):288–9. 

Aminoacids, including alpha-histidine, are included in the cell culture media formulation as they are essential nutrients for cell survival (see reference below).

Salazar A, Keusgen M, Von Hagen J. Amino acids in the cultivation of mammalian cells. Amino Acids.2016;48(5):1161–71. 

Therefore, a possible explanation is that copper and cobalt might interact with histidine forming a complex that inhibits the cobalt and copper individual effect. That would mean that they cannot inhibit, for example, the enzymes that hydroxylate HIF-1α and inhibit its translocation into the nucleus (cobalt effect) or favor the interaction of HIF-1a to the hypoxic response element (HRE) (copper effect). Therefore, no increase or enhancement of angiogenesis would be observed.

All this information has been added in the discussion of the paper (line 504-516):

“As we observed differences when both ions were cultured individually, these results point to the likelihood that both ions are mutually inhibiting or interacting with each other when being together. A possible hypothesis is that copper and cobalt might interact with some components of the cell culture media forming new complexes. In this regard, it is described in the literature that copper and cobalt can interact with each other with the presence of histidine, forming a new complex Co-Cu-His (42). As aminoacids, being histidine one of them, are added in the cell culture media formulations as essential nutrients (43), the possibility of forming the complex Co-Cu-His is possible. This would eventually explain why there is no increase in the angiogenesis response, as copper and cobalt might not be found freely to be able to enhance the HIF-1α binding to HRE (copper) and to inhibit the HIF-1α degradation (cobalt). However, future research is needed to test this hypothesis regarding the lack of synergy when both ions are combined together.”

MINOR COMMENTS

1. All fluorescence images should be marked as stained accordingly. Like actin filament (Green) / phalloidin (Green) and DAPI (Blue).

The reviewer is correct that the staining was not properly marked. We added more detail in figure captions of Figure 1, Figure 2 and Figure 7: “(actin filament staining (green); nucleus (blue)”.

2. Page 21, Line 432 & 433 “Stähli et al. reported a decreased HUVEC viability when 433 cultured with concentrations equivalent to 7, 21 and 63 μM compared to control (29).” Formatting should be corrected as per journal requirement.

We thank the reviewer for bringing up the citation style in this phrase. We noticed the same incorrect citation in another line and both have been corrected (line 430 and line 435).

---

## [Decision Letter · Decision Letter 1]

13 Oct 2021

Assessing the potential role of copper and cobalt in stimulating angiogenesis for tissue regeneration

PONE-D-21-25451R1

Dear Dr. Perez,

We’re pleased to inform you that your manuscript has been judged scientifically suitable for publication and will be formally accepted for publication once it meets all outstanding technical requirements.

Kind regards,

Selvaraj Vimalraj

Academic Editor

PLOS ONE

Additional Editor Comments (optional):

-

Reviewers' comments:

Reviewer's Responses to Questions

**Comments to the Author**

1. If the authors have adequately addressed your comments raised in a previous round of review and you feel that this manuscript is now acceptable for publication, you may indicate that here to bypass the “Comments to the Author” section, enter your conflict of interest statement in the “Confidential to Editor” section, and submit your "Accept" recommendation.

Reviewer #1: All comments have been addressed

Reviewer #2: All comments have been addressed

2. Is the manuscript technically sound, and do the data support the conclusions?

Reviewer #1: Yes

Reviewer #2: Partly

3. Has the statistical analysis been performed appropriately and rigorously? 

Reviewer #1: Yes

Reviewer #2: Yes

4. Have the authors made all data underlying the findings in their manuscript fully available?

Reviewer #1: Yes

Reviewer #2: Yes

5. Is the manuscript presented in an intelligible fashion and written in standard English?

Reviewer #1: Yes

Reviewer #2: Yes

6. Review Comments to the Author

Reviewer #1: (No Response)

Reviewer #2: The revised manuscript by Bosch-Rué et al. has addressed all my queries and suggestions to improve the manuscript.

I have no further comments.

7. PLOS authors have the option to publish the peer review history of their article (what does this mean?). If published, this will include your full peer review and any attached files.

Reviewer #1: No

Reviewer #2: No

---

## [Editor Report · Acceptance letter]

18 Oct 2021

PONE-D-21-25451R1 

Assessing the potential role of copper and cobalt in stimulating angiogenesis for tissue regeneration 

Dear Dr. Perez:

I'm pleased to inform you that your manuscript has been deemed suitable for publication in PLOS ONE. Congratulations! Your manuscript is now with our production department. 

Kind regards, 

on behalf of

Dr. Selvaraj Vimalraj 

Academic Editor

PLOS ONE